# Food insecurity and early childhood development among children 24–59 months in Nigeria: A multilevel mixed effects modelling of the social determinants of health inequities

Otobo I. Ujah[1,2]*, Omojo C. Adaji[3], Innocent A. O. Ujah[2], Russell S. Kirby[1]

1 College of Public Health, University of South Florida, Tampa, Florida, United States of America,
2 Department of Obstetrics and Gynaecology, Federal University of Health Sciences, Otukpo, Nigeria,
3 Sightsavers, Abuja, Nigeria

* otobo@usf.edu, otoboujah@yahoo.com

## Abstract

Food insecurity (FI) has been identified as a determinant of child development, yet evidence quantifying this association using the newly developed Early Childhood Development Index 2030 (ECDI2030) remains limited. Herein, we provide national estimates of early childhood development (ECD) risks using the ECDI2030 and examined to what extent FI was associated with ECD among children aged 24–59 months in Nigeria. This population based cross-sectional analyses used data from the UNICEF-supported 2021 Multiple Indicator Cluster Survey in Nigeria. The analytic sample comprised children aged 24–59 months (weighted N = 12,112). We measured early childhood development for each child using the ECDI2030, measured across three domains: learning, psychosocial well-being and health. Food insecurity was assessed using the Food Insecurity Experience Scale (FIES), categorized as none/mild, moderate and severe. We fitted mixed-effects multilevel logistic regression models, with random intercepts, to estimate the odds of association between FI status and ECD. A total of 11,494 children aged 24–59 months (mean ± SD age, 43.4 ± 9.9 months), including 5,797 boys (50.2%) and 5,697 girls (49.8%), were included in the study. Approximately 46.4% of children were developmentally off track and about 76% of children lived in food-insecure households. The intercept-only model indicated significant variation in ECD prevalence across communities ($\tau_{00}$ = 0.94, intraclass correlation = 0.22, $p$ < 0.0001), suggesting nonignorable variability in ECD across communities. Adjusting for confounders, we observed no significant association between FI and ECD. However, increasing child's age and disability status appeared as significant risk factors for higher odds of children being developmentally off track. These findings highlight that while FI alone may not explain ECD, a combination of individual and contextual factors plays a crucial role. Future interventions addressing ECD in Nigeria should consider these multidimensional influences to promote optimal child development.

**Data Availability Statement:** All data used in this study can be downloaded from https://mics.unicef.org/.

**Funding:** The author(s) received no specific funding for this work.

**Competing interests:** The authors have declared that no competing interests exist

## Introduction

Food insecurity (FI), characterized by limitations or uncertainties in accessing food in adequate quantity and quality or inability to access food in socially acceptable ways, is a growing and pervasive public health crisis [1–4]. According to a recent report by the United Nations Food and Agriculture Organization (UN FAO), approximately 2.4 billion people globally experienced moderate or severe FI in 2022 [5]. Food insecurity is particularly acute and persistent in Sub-Saharan Africa (SSA), where, in 2022, nearly two-thirds of the population experienced moderate to severe FI, and one in four experienced severe FI [5].

There is a growing concern about the detrimental impacts of FI on child performance and development [6–10]. For instance, a study in Canada found that children from households experiencing very low food security had 35% and 38% lower odds of meeting expectations in reading and mathematics, respectively [11]. Furthermore, FI is intricately linked to poverty and malnutrition [9]. Evidence shows that approximately 250 million children under 5 years of age globally were not developmentally "on track" due to extreme poverty and stunting [12]. Notably, in 2010, Nigeria, Ethiopia, DR Congo, and Tanzania were among the top 10 countries in SSA with the highest prevalence (> 60%) of children at risk of poor development [12]. More recently, a study in Ecuador found that preschool children experiencing marginal and moderate-severe FI had a 29% and 30% higher prevalence of reporting overall developmental delay, respectively [9]. This association varied across ECD domains, including literacy-numeracy, social-emotional, physical and cognitive development.

Several plausible interconnected pathways linking FI and ECD have been described in existing literature [13–16]. First, FI may result in poor dietary quality, leading to impaired cognitive and psychosocial well-being. Second, FI within contexts of financial hardships can result in limited investment in early childhood education resources, hindering child development. Furthermore, food-insecure caregivers experience higher stress levels, thereby reducing the quality and quantity of interactions with children. Lastly, children experiencing FI may demonstrate signs of psychological distress, lowering interaction quality with parents, teachers, and peers.

Early childhood development forms the foundation for the development of cognitive, motor, and social-emotional skills and, thus, remains central to the ability of individuals to survive, thrive and flourish during adulthood [12, 17, 18]. Public health interventions are urgently needed to address FI in SSA and its implications for early childhood development (ECD). However, there is currently a paucity of robust evidence quantifying the impact of FI on ECD in SSA [13], particularly in Nigeria. The lack of such evidence precludes the design and implementation of programs and policies needed to support national efforts in attaining the benchmarks set in the Sustainable Development Goals 2.1 and 4.2 [5]. Moreover, following the recent launch of the early child development index 2030 (ECDI2030) and its implementation in population-based surveys [19], there is a need to extend current evidence as it relates to the relationship of ECD with FI.

Herein, our objective was to estimate the prevalence of developmental risks and investigate the associations of early childhood development, based on the ECDI2030, with food insecurity and the social determinants of health. Framed by the socio-ecological model [20], we aimed to address the following specific research questions:

1. Is FI associated with ECD among children aged 24–59 months in Nigeria, after adjusting for socio-ecological factors?

2. What are the key socio-ecological factors associated with ECD among children aged 24–59 months in Nigeria?

3. How much variation in ECD among children aged 24–59 months is attributable to socio-ecological factors in Nigeria?

Addressing these questions extends our understanding of how FI and socio-ecological factors influence childhood development. This knowledge can inform interventions like early stimulation, responsive care, and nutritional support for children at risk of developmental delays due to FI [21]. In addition, considering that an important indicator for SDG 4.2 is measuring developmental progress in health, learning, and psychosocial well-being specifically among children aged 24–59 months [22], the present study focused on children within this age group.

## Theoretical framework

From a theoretical perspective, ECD, like other health behaviors and outcomes, occurs within an ecological framework that is influenced by a complex interplay of individual, family and broader contextual variables. Therefore, we base our study on Bronfenbrenner's Ecological Systems Theory (EST), which informs both its conceptual and empirical aspects. According to EST, human development occurs within a complex set of relationships influenced by multiple interacting environmental factors [20]. In the context of ECD, these factors occur across multiple levels: the microsystem (age, sex, race/ethnicity, education), the mesosystem (parental education, religion, household income), the exosystem (community socioeconomic status, availability of early childhood education programs), and the macrosystem (government policies and legislation) [23–25]. The EST has been used across a diverse range of health behaviors and outcomes including sexual behaviors [26], mental health [27], food insecurity [28], physical activity [29] and sleep [30]. For this study, the EST provides a compelling lens for understanding ECD by accounting for the intersecting influences of individual, familial, community, and societal factors. This framework not only informs our analytic strategy but also aids in interpreting findings and designing targeted public health interventions aimed at optimizing development in early childhood.

## Methods

### Study design and data source

In this cross-sectional study, we performed a secondary analysis of nationally representative data drawn from Round 6 of the Multiple Indicator Cluster Survey (MICS6), conducted in 2021 in Nigeria, sponsored by UNICEF and implemented by the National Bureau of Statistics (NBS). The MICS is a population-based household survey that collects nationally representative data on social and health indicators in low- and middle-income countries (LMICs) from representative samples of households, men, women and children. The survey employed a multistage stratified cluster design.

In the first stage, 2,076 primary sampling units (PSUs) were randomly selected from the 2006 Population and Housing Census of the Federal Republic of Nigeria (NPHC). In the second stage, 20 households were randomly selected from each PSU using a probability proportional-to-size sampling. Data collection was performed using Computer-Assisted Personal Interviewing (CAPI) technology, with face-to-face interviews conducted in respondents' households. Documentation regarding the Nigerian MICS sampling design and data collection techniques can be found elsewhere [31]. For this study, we merged the child dataset, which contained variables related to ECD outcomes, with the household dataset, which contains data on FI. The datasets used in this study are publicly available from the MICS website (https://mics.unicef.org/).

## Ethical considerations

The MICS protocol was approved by the National Bureau of Statistics (NBS) and UNICEF. According to the 2021 MICS6 report [31], all participants provided verbal consent, including minors aged 15–17 years, who required prior adult consent. Participants were informed about the voluntary nature of participation, data confidentiality, anonymity, and their right to refuse questions or terminate the interview. Although this study involves human participants, IRB review and approval were not required as it is based on an analysis of secondary data, and prior ethical approval had been obtained by the primary data collectors.

## Analytic sample

Among all children under 5 years of age participating in MICS6, all boys and girls aged 24–59 months were eligible for inclusion in the analysis (19,463/33,103). We further restricted the analysis to participants with complete and valid data on socio-demographic characteristics, maternal characteristics, overall ECD2030 scores, and food insecurity (FI). As a result, the final study sample consisted of 11,494 children aged 29–59 months (weighted N = 12,112), belonging to 9,539 households nested within 1,718 clusters (Fig 1). The respondents providing information for these children were either their mothers or caregivers.

**Outcome assessment: Early Childhood Development (ECD).** We assessed ECD using the 20-item Early Childhood Development Index (ECDI) 2030. Launched in 2020, the ECDI2030 is a comprehensive caregiver-reported tool used to estimate the percentage of children aged 24–59 months who are developmentally on track relative to their age, across the domains of health, learning and psychosocial well-being [19]. The ECDI2030 consists of 20 questions, each designed to measure specific developmental constructs within these domains (Table 1). The ECDI2030 was recently incorporated into Nigeria's MICS6 in 2021, serving as a key tool for monitoring progress on SDG proxy indicator 4.2.1 –"proportion of children aged 24–59 months who are developmentally on track in health, learning, and psychosocial well-being, by sex". Further details on the ECDI2030 are published elsewhere [19]. Based on the age-specific ECDI summary score, we created a two-category variable to classify children as developmentally 'on track' or 'off track' in accordance with the ECDI2030 guidelines.

**Exposure assessment: Food insecurity.** The primary independent variable in this study was household food insecurity (FI) over the past 12 months, assessed using the widely validated 8-item Household Food Insecurity Experience Scale (FIES) developed by the United Nations Food and Agriculture Organization (UN FAO). The FIES is a tool used to estimate the prevalence of moderate or severe food insecurity within populations, in alignment with Sustainable Development Goal (SDG) indicator 2.1.2. Participants were asked about their household's food insecurity status over the preceding 12 months. The questions addressed whether there was a time when they or members of their household could not afford healthy and nutritious food due to financial constraints, if their household experienced food shortages due to limited financial resources, or if they or others in the household went without eating for an entire day due to financial constraints (Table 2). Response options for each question included "Yes", "No" or "Don't know". Composite household food insecurity scores were derived by summing the affirmative responses, yielding a score range from 0–8. In this study, FI was categorized based on the FIES score as none/mild, moderate or severe.

## Confounding variables

The covariates used in this study were defined *a priori* based on previous studies [6, 7, 10, 32], selected based on their availability in the dataset, and organized according to three levels of the EST into: microsystem, mesosystem and exosystem. The microsystem variables included

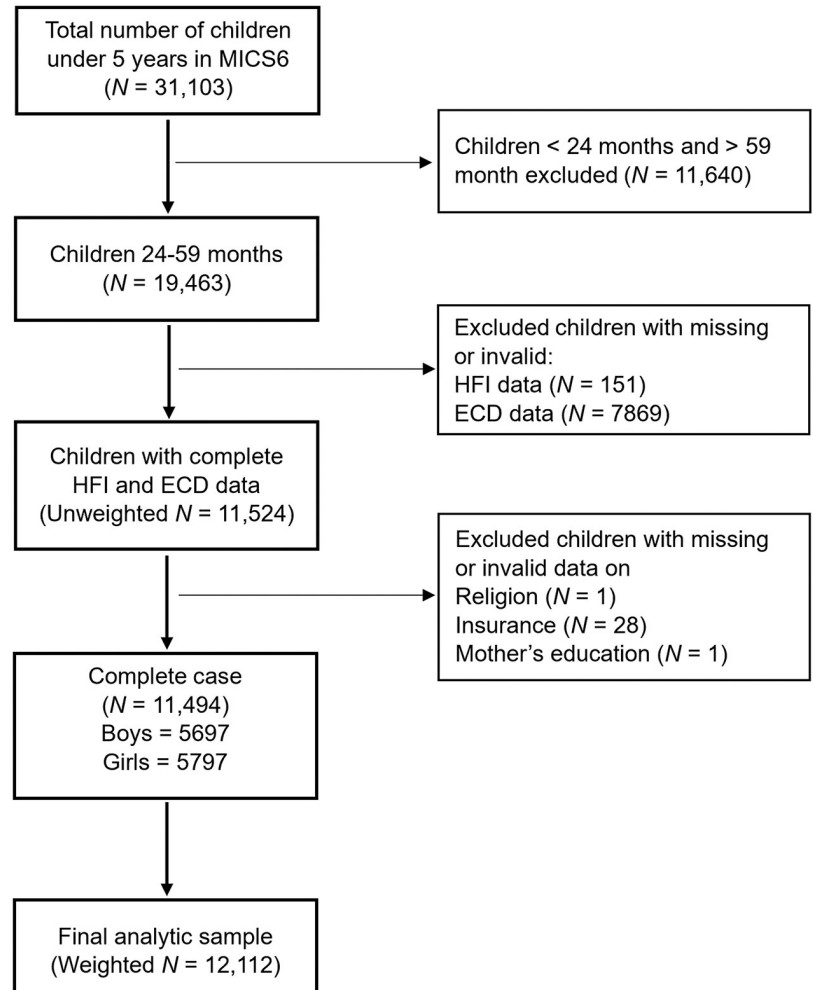

**Fig 1. Flow chart depicting the selection of study participants for a study of the association between food insecurity and early children development (ECD) among children 24–59 months in Nigeria, Multiple Indicator Cluster Survey (MICS6), 2021.**

child-level characteristics such as age in months (24–29, 30–35, 36–41, 42–47, 48–59), sex (boy or girl), disability status (no functional difficulty, functional difficulty), and health insurance coverage (not covered, covered). Mesosystem variables included the number of children in the household (1, >2), religious affiliation (Christian or Non-Christian), mother's educational attainment (less than secondary, secondary, and higher than secondary), and household wealth quintile (1, 2, 3, 4 or 5). Exosystem-level or variables included place of residence (rural or urban) and geographic region of residence (North Central, North East, North West, South East, South South or South West).

## Statistical analysis

Data analyses were performed using SAS version 9.4 (SAS Institute Inc., Cary, NC) and R software (version 4.3.0). We applied sampling weights to the survey data using the weights assigned to each child in the Nigeria MICS6 survey. We used the SAS PROC SURVEY

**Table 1. Domains, items and scoring of the ECDI2030 used in MICS6, Nigeria, 2021.**

| Domain | Items | Scoring |
|---|---|---|
| **Learning** | Can (name) say at least ten or more words like "Mama" or "ball"? | 1 |
| | Can (name) speak using sentences of three or more words that go together, for example, "I want water" or "The house is big"? | 1 |
| | Can (name) speak using sentences of five or more words that go together? | 1 |
| | If you show (name) an object he/she knows well, such as a cup or animal, can he/she consistently name it? | 1 |
| | Can (name) correctly use any of the words 'I,' 'you,' 'she,' or 'he', for example, "I go to the store," or "He eats rice"? | 1 |
| | Can (name) recognize at least five letters of the alphabet? | 1 |
| | If you ask (name) to give you three objects, such as three stones or three beans, does (he/she) give you the correct amount? | 1 |
| | Can (name) count 10 objects, for example, 10 fingers or blocks, without mistakes? | 1 |
| | Does (name) know all numbers from 1 to 5? | 1 |
| | Can (name) write his/her own name? | 1 |
| | Can (name) do an activity such as colouring without repeatedly asking for help or giving up too quickly? | 1 |
| **Psychosocial well-being** | Does (name) get along well with other children? | 1 |
| | Does (name) ask about familiar people other than parents when they are not there, for example, "Where is Grandma?"? | 1 |
| | Does (name) offer to help someone who seems to need help? | 1 |
| | How often does (name) seem to be very sad or depressed? | 1 |
| | Compared with children of the same age, does (name) kick, bite or hit other children or adults? | 1 |
| **Health** | Can (name) dress him/herself, that is, put on pants and shirt without help? | 1 |
| | Can the child fasten and unfasten buttons without help? | 1 |
| | Can (name) jump up with both feet leaving the ground? | 1 |
| | Can (name) walk on an uneven surface, for example, a bumpy or steep road, without falling? | 1 |

**Adapted Early Childhood Development Index 2030 (ECDI2030)**: Percentage of children aged 24 to 59 months who have achieved the minimum number of milestones expected for their age group (Children aged 24–29 months: 7 of the 20 items; Children aged 30–35 months: 9 of the 20 items; Children aged 36–41 months: 11 of the 20 items; Children aged 42–47 months: 13 of the 20 items; Children aged 48–59 months: 15 of the 20 item)

**Source**: The Early Childhood Development Index 2030: A New Measure of Early Childhood development, https://data.unicef.org/resources/early-childhood-development-index-2030-ecdi2030/

commands to apply these weights by incorporating the effects of weights, clustering and stratification to calculate means and standard errors (SE), as well as frequencies and percentages (%). We further examined between-group differences for categorical variables using the Rao-Scott Chi-squared test of association.

**Multilevel model building strategy.** Given the hierarchical structure of the data from the MICS, with children ($i$) nested within clusters ($j$), and considering the dichotomous nature of the outcome variable (reporting developmentally on track or not), we specified several two-level logistic regression models with random intercepts. These models were adjusted for microsystem, mesosystem and exosystem-level characteristics. We used SAS PROC GLIMMIX with a binomial distribution and the LOGIT link function. All models in this study were estimated using the maximum likelihood approach (METHOD = LAPLACE). We also employed the CONTAINMENT approximation option (DDFM = CONTAIN) to estimate the degrees of freedom for the fixed effects.

**Table 2. Description Food Insecurity Experience Scale (FIES) used in the Multiple Indicator Cluster Survey (MICS), 2021.**

| FIES Indicators | Short Reference | Description |
|---|---|---|
| Q1 | WORRIED | During the last 1 year, was there a time when you or others in your household worried about not having enough food to eat because of a lack of money or other resources? |
| Q2 | HEALTHY | During the last 1 year, was there a time when you or others in your household were unable to eat healthy and nutritious food because of a lack of money or other resources? |
| Q3 | FEWFOODS | During the last 1 year, was there a time when you or others in your household ate only a few kinds of foods because of a lack of money or other resources? |
| Q4 | SKIPPED | During the last 1 year, was there a time when you or others in your household had to skip a meal because there was not enough money or other resources to get food? |
| Q5 | ATELESS | During the last 1 year, was there a time when you or others in your household ate less than you thought you should because of a lack of money or other resources? |
| Q6 | RANOUT | During the last 1 year, was there a time when your household ran out of food because of a lack of money or other resources? |
| Q7 | HUNGRY | During the last 1 year, was there a time when you or others in your household were hungry but did not eat because there was not enough money or other resources for food? |
| Q8 | WHLDAY | During the last 1 year, was there a time when you or others in your household went without eating for a whole day because of a lack of money or other resources? |

Our modeling approach began by specifying a null (unconditional) model, excluding any predictors, to assess between-cluster variation in ECD and to determine the appropriateness of a multilevel modeling approach. Subsequently, we developed more complex conditional models by sequentially including predictors. Model I included the main exposure variable–FI; Model II included Model I and adjusted for microsystem-level factors, while Model III incorporated Model II and adjusted for mesosystem-level variables. Lastly, Model IV was the fully adjusted model, which included Model III with adjustments for exosystem-level factors. Fixed effects are represented as crude and adjusted odds ratios (ORs) along with their corresponding 95% confidence intervals (CIs). All tests were two-tailed, and $p$-values $< 0.05$ were considered statistically significant.

We assessed random effects for ECD using three standard metrics: intraclass correlation coefficient (ICC), Median Odds Ratio (MOR), and Proportional Change in Variance (PCV). The ICC quantifies the proportion of total observed variability in ECD status that can be attributed to between-cluster variability. The MOR estimates the variability between clusters by comparing two individuals randomly selected from different clusters, while the PCV estimates the variation explained by the multilevel models. To calculate the PCV, the $\tau_{00}$ value for conditional models (Models I-IV) were compared to that of the previous model: $[\tau_{00(n-1)} — \tau_{00(n-2)}/\tau_{00(n-1)}]$, where $\tau_{00}$ represents the between-cluster variability. To assess the goodness of fit of the different models, we used the Akaike Information Criterion (AIC). Smaller AIC values indicated better-fitting models.

## Results

### Sample characteristics

Data were analyzed from a sample of 11,494 children (weighted to represent 12,112 children aged 24–59 months). The average age was 43.4 ± 9.9 months for the total sample, 43.2 ± 10.0

months for boys, and 43.5 ± 9.7 months for girls. The microsystem, mesosystem and exosystem-level characteristics of the study participants are presented in Table 3. Approximately 49.8% of the sample were girls, while 50.2% were boys. Nearly two-thirds of the children were between the ages of 24–48 months. Most children (> 95%) were not covered by health insurance and did not have a functional difficulty.

Among the mothers of children aged 24–59 months included in the study, slightly more than half (55.7%) had less than secondary education. At least two out of three children resided in households with 2 or more children under 5 years of age (66.1%) and in households whose religious affiliation was Christianity (60.3%). About 44% of children resided in households with a poor wealth index. About two-thirds (62.5%) of the children resided in households located in the rural areas while 34.7% resided in households located in the North West region. A total of 2,629 children (23.5%) were in food secure households, while 3,567 (31.9%) and 5,298 (44.6%) were living in households experiencing moderate and severe FI, respectively.

In our sample, the mean ECDI2030 score was 11.98 (SD = 3.54; range = 2–20). Based on individual ECD domain, the mean learning score was 5.77 (SD = 2.25; range = 1–11), the mean health score was 2.53 (SD = 1.18; range = 0–4) while the mean psychosocial well-being score was 3.68 (SD = 1.01; range = 1–5). For children who were developmentally on track, the mean ECDI2030 score was 13.77 (SD = 3.31) and 10.04 (SD = 2.65) for children who were developmentally off track.

Table 3 also shows the univariate associations between ECD and the characteristics of the study participants. Approximately 6,498 (53.6%) children were developmentally on track while 5,614 (46.4%) were developmentally off track ($p < 0.0001$). Children who were developmentally on track were more likely to live in food secure households, while those who were not developmentally on track were slightly more likely to live in households experiencing FI (moderate or severe), as shown in Table 3. Fig 2 illustrates the weighted prevalence of affirmative responses to each item of the FIES by ECD status among children in the sample. There were statistically significant differences between the ECD categories and each item of the FIES ($p < 0.05$) except for "ATELESS", "RANOUT" and "HUNGRY". Additionally, Fig 3 shows the prevalence of ECD status based on FI status among children aged 24–59 months in Nigeria.

All the confounding variables demonstrated differential distribution with respect to the outcome variable, early childhood development. As shown in Table 3, compared to children who were developmentally on track, those who were developmentally off track were more likely to be older (52.2% versus 31.2%, $p < 0.001$), experience functional difficulties (3.8% versus 1.6%, $p < 0.001$), have mothers with less than a secondary level of education (70.68% versus 49.2%, $p < 0.001$), reside in households with two or more children under 5 years of age (71.6% versus 61.4%, $p < 0.001$), be affiliated with a religion other than Christianity (72.2% versus 50.0%, $p < 0.001$), reside in poor households (59.9% versus 33.0%, $p < 0.001$), live in rural areas (74.6% versus 51.9%, $p < 0.001$) and in the Northern region (78.2% versus 51.1%, $p < 0.001$).

## Multilevel analysis

**Measures of association (fixed effects).** The estimated intercept for the empty model was -0.22 (Table 4), suggesting that in a typical cluster, where the random effect on the logit scale is zero, the odds of children aged 24–59 months being developmentally off track were 0.80, corresponding to a probability of 0.44. Table 4 presents the results of the multilevel logistic regression analysis for FI and ECD. The crude model (Model I) shows that increasing FI status was associated with increasingly higher odds of being developmentally off track. Specifically, children who lived in households experiencing moderate FI vs No FI had a 22% higher odds of

**Table 3. Characteristics among children aged 24–59 months old by early childhood development status, Multiple Indicator Cluster Survey, 2021 (N = 11494).**

| Characteristics | Overall | | Early Childhood Development (ECD) status | | | | |
|---|---|---|---|---|---|---|---|
| | | | On Track *N* = 5,969 53.6% | | Off Track *N* = 5,525 46.4% | | *p* value |
| | *N* | % | *N* | % | *N* | % | |
| **Household food insecurity** | | | | | | | |
| None | 2,629 | 23.5 | 1,626 | 25.0 | 1,175 | 21.7 | **0.0297** |
| Moderate | 3,567 | 31.9 | 2,038 | 31.4 | 1,827 | 32.5 | |
| Severe | 5,298 | 44.6 | 2,834 | 43.6 | 2,569 | 45.8 | |
| **Sex** | | | | | | | |
| Boy | 5,797 | 50.2 | 3,226 | 49.6 | 2,848 | 50.7 | 0.43 |
| Girl | 5,697 | 49.8 | 3,272 | 50.4 | 2,766 | 49.3 | |
| **Age, mean (SD), months** | 43.4 (9.9) | | 41.1 (10.4) | | 45.8 (8.7) | | |
| **Age category, months** | | | | | | | |
| 24–29 | 1,381 | 12.3 | 11,16 | 19.1 | 265 | 4.4 | **<0.0001** |
| 30–35 | 1,308 | 11.5 | 855 | 14.4 | 453 | 8.2 | |
| 36–41 | 2,208 | 19.9 | 1,277 | 21.9 | 931 | 17.4 | |
| 42–47 | 1,759 | 15.5 | 745 | 13.3 | 1,014 | 17.9 | |
| 48–59 | 4,838 | 40.9 | 1,976 | 31.2 | 2,862 | 52.2 | |
| **Disability status** | | | | | | | |
| No functional difficulty | 11,188 | 97.4 | 5,875 | 98.4 | 5,313 | 96.2 | **<0.0001** |
| Functional difficulty | 306 | 2.6 | 94 | 1.6 | 212 | 3.8 | |
| **Health insurance coverage** | | | | | | | |
| No | 11,177 | 96.8 | 5,728 | 95.2 | 5,449 | 98.5 | **<0.0001** |
| Yes | 317 | 3.2 | 241 | 4.8 | 76 | 1.5 | |
| **Mother's education level** | | | | | | | |
| Less than secondary | 6,707 | 55.7 | 2,748 | 42.9 | 3,959 | 70.6 | **<0.0001** |
| Secondary | 3,707 | 32.6 | 2,375 | 39.7 | 1,332 | 24.4 | |
| Higher secondary | 1,080 | 11.7 | 846 | 17.4 | 234 | 5.0 | |
| **Number of children < 5 years in household** | | | | | | | |
| 1 | 3,718 | 33.9 | 2,191 | 38.6 | 1,527 | 28.4 | **<0.0001** |
| 2 or more | 7,776 | 66.1 | 3,778 | 61.4 | 3,998 | 71.6 | |
| **Religious affiliation** | | | | | | | |
| Christianity | 4,608 | 39.7 | 2,923 | 50.0 | 1,685 | 27.8 | **<0.0001** |
| Other | 6,886 | 60.3 | 3,046 | 50.0 | 3,840 | 72.2 | |
| **Household wealth index** | | | | | | | |
| Poorest | 3,112 | 23.8 | 1,149 | 15.5 | 1,963 | 33.4 | **<0.0001** |
| Poorer | 2,766 | 21.7 | 1,246 | 17.5 | 1,520 | 26.5 | |
| Middle | 2,443 | 19.9 | 1,332 | 19.9 | 1,111 | 19.9 | |
| Richer | 1,837 | 17.6 | 1,169 | 21.8 | 668 | 12.8 | |
| Richest | 1,336 | 17.0 | 1,073 | 25.3 | 263 | 7.4 | |
| **Place of residence** | | | | | | | |
| Rural | 8,190 | 62.5 | 3,817 | 51.9 | 4,373 | 74.6 | **<0.0001** |
| Urban | 3,304 | 37.5 | 2,152 | 48.0 | 1,152 | 25.4 | |
| **Geographic region** | | | | | | | |
| North Central | 2,144 | 12.9 | 1,079 | 12.3 | 1,065 | 13.7 | **<0.0001** |
| North East | 2,799 | 16.1 | 1,305 | 12.9 | 1,494 | 19.7 | |
| North West | 3,119 | 34.7 | 1,215 | 25.9 | 1,904 | 44.8 | |
| South East | 1,082 | 10.1 | 706 | 12.7 | 376 | 7.0 | |

*(Continued)*

**Table 3.** (Continued)

| Characteristics | Overall | | Early Childhood Development (ECD) status | | | | |
|---|---|---|---|---|---|---|---|
| | | | On Track $N$ = 5,969 53.6% | | Off Track $N$ = 5,525 46.4% | | $p$ value |
| | $N$ | % | $N$ | % | $N$ | % | |
| South South | 1,183 | 11.3 | 786 | 14.8 | 397 | 7.2 | |
| South West | 1,167 | 14.9 | 878 | 21.3 | 289 | 7.5 | |
| Number of observations (unweighted sample) | 11,494 | | | | | | |
| Population size (weighted sample) | 12,112 | | | | | | |
| No. of primary sampling units (PSUs) | 1,718 | | | | | | |
| No. of strata | 37 | | | | | | |

*Notes*: Frequencies are unweighted; Percentages (%) are weighted to adjusted for the complex sampling design of the survey. Percentages may not total 100 due to rounding

being developmentally off track (OR = 1.22; 95% CI: 1.07–1.39) while those who lived in households experiencing severe FI vs No FI had a 26% higher odds of being developmentally off track (OR = 1.26; 95% CI: 1.11–1.42). However, after accounting for microsystem, mesosystem and exosystem level factors (Table 4, Model IV), these associations between FI status and ECD status were no longer statistically significant.

From Table 4, Model IV (the fully adjusted model) was the model which had the best fit for the data based on the AIC and BIC values (13,383.82 and 13520.04, respectively), and therefore was used to answer the remaining research questions. In the final model which fully adjusted

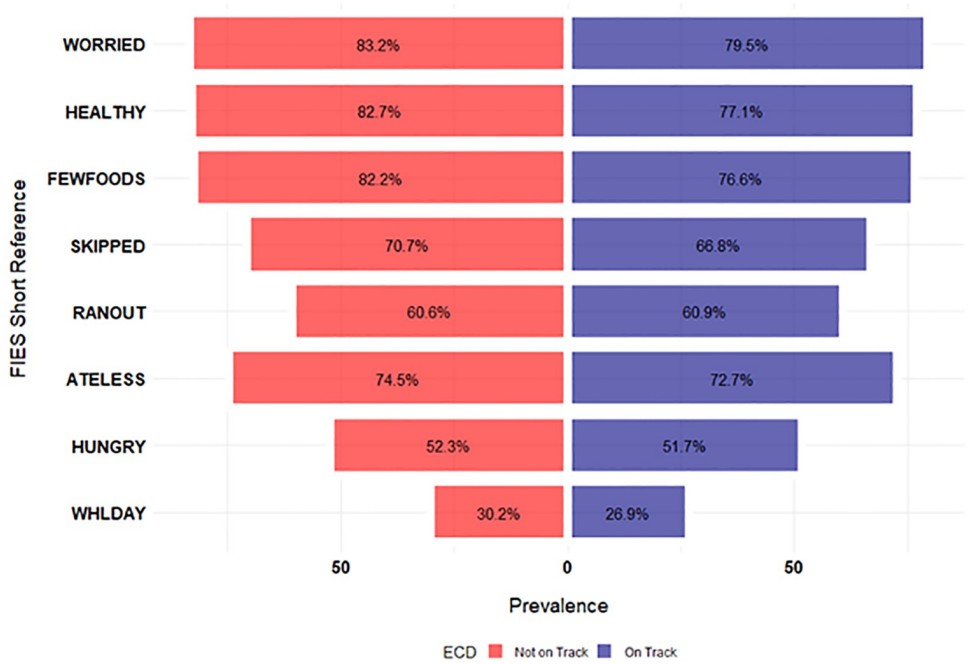

**Fig 2. Single items of the FIES among children 24–59 months who were developmentally on track and not on track.**

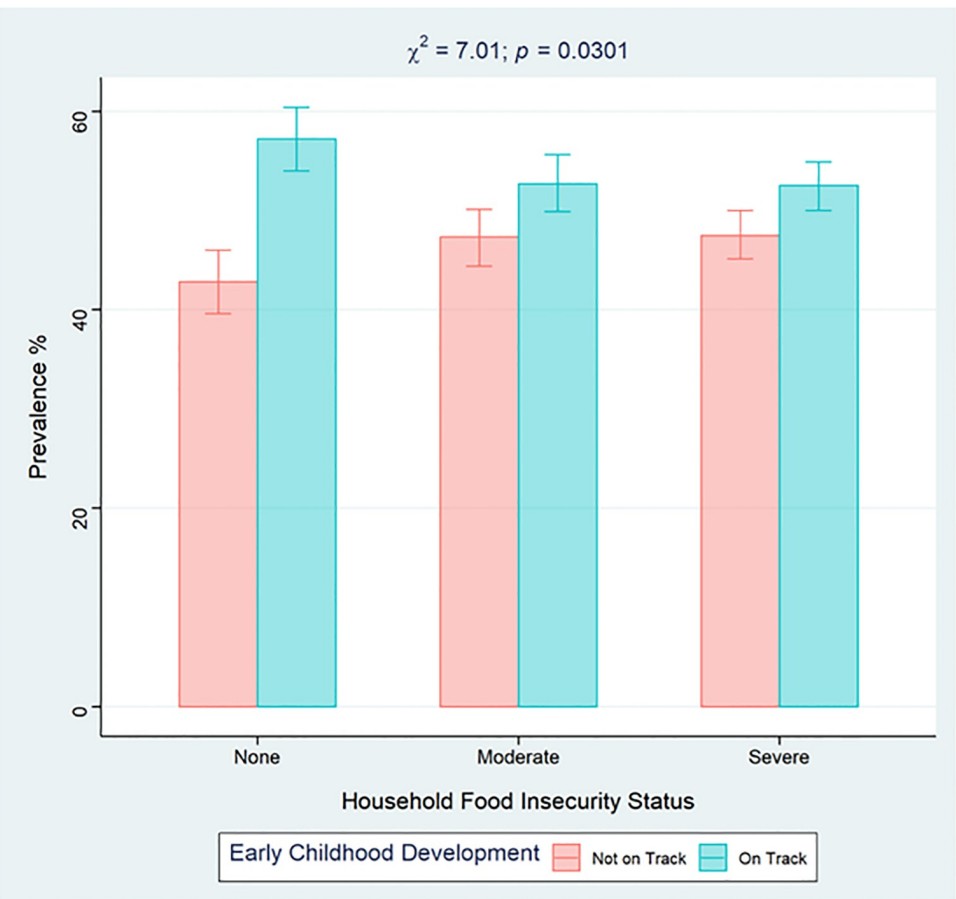

**Fig 3. Weighted prevalence of early childhood development status according to household food insecurity status among children 24–59 months in Nigeria.** Error bars represent 95% confidence intervals.

for the effects of microsystem, mesosystem and exosystem characteristics, child's age, disability status, mother's level of educational attainment, religious affiliation, household wealth index, place of residence and region of residence were significantly associated with the odds of children being developmentally off track. At the microsystem level, the odds of being developmentally off track increased with each age category. Specifically, the odds were 2.78 times higher among those aged 30–35 months (aOR = 2.78, 95% CI = 2.25–3.43), 3.47 times higher among those aged 36–41 months (aOR = 3.47, 95% CI = 2.85–4.22), 8.38 times higher among those aged 42–47 months (aOR = 8.38, 95% CI = 6.84–10.27), and 8.28 times higher among those aged 48–59 months (aOR = 8.28, 95% CI = 6.90–9.94). Furthermore, children who had a functional difficulty were 2.5 times (aOR = 2.54; 95% CI = 1.88–3.43) more likely to be developmentally off track compared to children with no functional difficulty.

At the mesosystem level, children born to mothers with secondary education had a 33% lower likelihood of being developmentally off track compared to those born to mothers with less than secondary education (aOR = 0.67; 95% CI = 0.60–0.76). Additionally, children born to mothers with higher than secondary education had a 55% lower likelihood of being developmentally off track than those born to mothers with less than secondary education (aOR = 0.45; 95% CI = 0.37–0.56). Furthermore, children residing in households affiliated with Christianity

**Table 4. Results from the two-level logistic regression models investigating the association between household food insecurity and development in early childhood, adjusting for child, maternal/family and contextual level factors among children 24–59 months in Nigeria, N = 11494.**

Outcome variable: Early Childhood Development (Reference = On Track)

| Variables | Null Model[a] | | Model I[b] | | Model II[c] | | Model III[d] | | Model IV[e] | |
|---|---|---|---|---|---|---|---|---|---|---|
| | Odds | | OR | 95% CI | aOR | 95% CI | aOR | 95% CI | aOR | 95% CI |
| **Fixed effects** | | | | | | | | | | |
| Intercept[†] | **0.80** | **0.75–0.85***** | **0.68** | **0.60–0.76***** | **0.14** | **0.11–0.17***** | **0.41** | **0.33–0.51***** | **0.43** | **0.34–0.55***** |
| Food insecurity status | | | | | | | | | | |
| None | | | 1.00 | | 1.00 | | 1.00 | | 1.00 | |
| Moderate FI | | | **1.22** | **1.07–1.39**** | **1.18** | **1.03–1.35*** | 1.04 | 0.91–1.19 | 1.05 | 0.91–1.20 |
| Severe FI | | | **1.26** | **1.11–1.42**** | **1.22** | **1.07–1.39**** | 0.98 | 0.86–1.12 | 1.00 | 0.88–1.14 |
| **Child characteristics** | | | | | | | | | | |
| Age, months | | | | | | | | | | |
| 24–29 | | | | | 1.00 | | 1.00 | | 1.00 | |
| 30–35 | | | | | **2.66** | **2.15–3.27***** | **2.74** | **2.22–3.37***** | **2.78** | **2.25–3.43***** |
| 36–41 | | | | | **3.61** | **2.96–4.41***** | **3.45** | **2.83–4.20***** | **3.47** | **2.85–4.22***** |
| 42–47 | | | | | **8.12** | **6.62–9.98***** | **8.29** | **6.77–10.15***** | **8.38** | **6.84–10.27***** |
| 48–59 | | | | | **8.49** | **7.06–10.21***** | **8.12** | **6.77–9.75***** | **8.28** | **6.90–9.94***** |
| Sex | | | | | | | | | | |
| Boy | | | | | 1.00 | | 1.00 | | 1.00 | |
| Girl | | | | | 0.97 | 0.89–1.05 | 0.97 | 0.89–1.06 | 0.97 | 0.89–1.06 |
| Disability status | | | | | | | | | | |
| No functional difficulty | | | | | 1.00 | | 1.00 | | 1.00 | |
| Functional difficulty | | | | | **3.08** | **2.27–4.18***** | **2.66** | **1.97–3.59***** | **2.54** | **1.88–3.43***** |
| Health insurance coverage | | | | | | | | | | |
| No | | | | | **1.00** | | 1.00 | | 1.00 | |
| Yes | | | | | **0.38** | **0.27–0.53***** | 0.80 | 0.56–1.13 | 0.80 | 0.57–1.12 |
| **Maternal/family characteristics** | | | | | | | | | | |
| Mother's education level | | | | | | | | | | |
| Less than secondary | | | | | | | 1.00 | | 1.00 | |
| Secondary | | | | | | | **0.64** | **0.57–0.72***** | **0.67** | **0.60–0.76***** |
| Higher secondary | | | | | | | **0.44** | **0.36–0.54***** | **0.45** | **0.37–0.56***** |
| Number of children < 5 years in household | | | | | | | | | | |
| 1 | | | | | | | 1.00 | | 1.00 | |
| ≥ 2 | | | | | | | **1.11** | **1.01–1.23*** | 1.07 | 0.96–1.18 |
| Religious affiliation | | | | | | | | | | |
| Other | | | | | | | 1.00 | | 1.00 | |
| Christian | | | | | | | **0.60** | **0.54–0.68***** | **0.74** | **0.64–0.86***** |
| Household wealth index | | | | | | | | | | |
| 1 (Poorest) | | | | | | | 1.00 | | 1.00 | |
| 2 | | | | | | | **0.81** | **0.71–0.92**** | **0.84** | **0.74–0.96*** |
| 3 | | | | | | | **0.58** | **0.50–0.67***** | **0.67** | **0.58–0.78***** |
| 4 | | | | | | | **0.43** | **0.36–0.51***** | **0.54** | **0.45–0.65***** |
| 5 (Richest) | | | | | | | **0.23** | **0.18–0.28***** | **0.31** | **0.25–0.39***** |
| **Contextual level characteristics** | | | | | | | | | | |
| Place of residence | | | | | | | | | | |
| Rural | | | | | | | | | 1.00 | |
| Urban | | | | | | | | | **0.80** | **0.69–0.92**** |
| Geographic region | | | | | | | | | | |

*(Continued)*

**Table 4.** (Continued)

| Outcome variable: Early Childhood Development (Reference = On Track) | | | | | | | | | |
|---|---|---|---|---|---|---|---|---|---|
| **Variables** | **Null Model[a]** | | **Model I[b]** | | **Model II[c]** | | **Model III[d]** | | **Model IV[e]** |
| | **Odds** | | **OR** | **95% CI** | **aOR** | **95% CI** | **aOR** | **95% CI** | **aOR** | **95% CI** |
| North Central | | | | | | | | | 1.00 | |
| North East | | | | | | | | | **0.84** | **0.71–0.99**[*] |
| North West | | | | | | | | | 1.11 | 0.94–1.31 |
| South East | | | | | | | | | **0.67** | **0.53–0.85**[**] |
| South South | | | | | | | | | **0.58** | **0.47–0.72**[***] |
| South West | | | | | | | | | **0.45** | **0.36–0.56**[***] |

*Notes*: Estimation method = Maximum likelihood; Containment degrees of freedom; All estimates are weighted for the survey's complex sampling design. Boldface indicates statistically significant results at the 0·05 level.

Abbreviations: OR–odds ratio, aOR–adjusted odds ratio CI–confidence interval.

[a] Null model unconditional model, baseline model without any predictor variables

[b] Model I–includes the main explanatory variable (FI)

[c] Model II–Model I adjusted for only microsystem level characteristics

[d] Model III–Model II adjusted for only mesosytem level characteristics

[e] Model IV–Model III adjusted for exosystem level characteristics (full model)

[†] Estimates presented as odds

[***]$p < 0.001$,

[**]$p < 0.01$,

[*]$p < 0.05$

had 26% lower odds of being developmentally off-track compared to children in non-Christian households (aOR = 0.74; 95% CI = 0.64–0.86). The odds of being developmentally off-track decreased with increasing household wealth index. The odds of being developmentally off-track among children from poorer households were 16% lower (aOR = 0.84; 95% CI = 0.74–0.96), 33% lower for children from middle-income households (aOR = 0.67; 95% CI = 0.58–0.78), 46% lower for children from richer households (aOR = 0.54; 95% CI = 0.45–0.65), and 69% lower for children from the richest households (aOR = 0.31; 95% CI = 0.25–0.39), compared to children from the poorest households.

At the exosystem level, the odds of being developmentally off track were 20% lower for children residing in urban areas compared to their counterparts residing in rural areas (aOR = 0.80; 95% CI = 0.69–0.92). Furthermore, the odds of being developmentally off track differed by region. Children residing in the North East had 16% lower odds (adjusted odds ratio [aOR] = 0.84; 95% CI = 0.71–0.99), those in the South East had 33% lower odds (aOR = 0.67; 95% CI = 0.53–0.85), those in the South South had 42% lower odds (aOR = 0.58; 95% CI = 0.47–0.72), and those in the South West had 69% lower odds (aOR = 0.45, 95% CI = 0.36–0.56), all compared to children in the North Central.

**Measures of variation (random effects).** Table 5 presents estimates of the random effects from the multilevel analysis. The probability of children aged 24–59 months being developmentally off track varied significantly across clusters, as indicated by the random effect estimate [$\tau_{00}$ = 0.942, z(1,717) = 12.64, $p < 0.0001$]. Systematic differences across clusters accounted for approximately 22% of the variability in the odds of children being developmentally on track (ICC = 0.21), with the remaining 78% attributed to individual or other unexplained factors. The between-cluster variability declined across successive models, from 22% in the unconditional model to 22.6% in the microsystem level model, 10.9% in the mesosystem

**Table 5. Results from the random intercept model (measure of variation) for early childhood development at cluster level by multilevel logistic regression analysis.**

| Random effects | Null model | Model I | Model II | Model III | Model IV |
|---|---|---|---|---|---|
| Cluster-level variance (SE) | 0.94 (0.08) | 0.86 (0.07) | 0.96 (0.08) | 0.40 (0.05) | 0.37 (0.05) |
| *p* value | < 0.0001 | < 0.0001 | < 0.0001 | < 0.0001 | < 0.0001 |
| ICC (%) | 22.27 | 20.77 | 22.64 | 10.90 | 10.12 |
| MOR | 2.52 | 2.43 | 2.55 | 1.83 | 1.79 |
| PCV (%) | - | 8.48% | -2.19 | 57.31 | 60.69 |
| **Model fit statistics** | | | | | |
| AIC | 15242.15 | 15264.23 | 14191.51 | 13470.29 | 13383.82 |
| BIC | 15253.05 | 15286.03 | 14251.44 | 13573.82 | 13520.04 |

Abbreviations: ICC-Intraclass correlation coefficient, MOR–Median Odds Ratio, PCV–Proportional Change in Variance, AIC–Akaike Information Criteria; BIC–Bayesian Information Criteria

[a] Null model unconditional model, baseline model without any predictor variables

[b] Model I–includes the main explanatory variable (FI)

[c] Model II–Model I adjusted for only microsystem level characteristics

[d] Model III–Model II adjusted for only mesosystem level characteristics

[e] Model IV–Model III adjusted for exosystem level characteristics (full model)

level only model, and 10.1% in the fully adjusted model. In the null model, the MOR was estimated to be 2.5. This implies that children residing in a cluster characterized by being developmentally on track had 2.5 times higher odds of being developmentally on track compared to a child residing in a cluster where children were not developmentally on track. After including microsystem, mesosystem and exosystem level characteristics in the model, the MOR decreased to 1.79. This indicates that the effect of clustering remains statistically significant in the fully adjusted model. Notably, the PCV indicated that the addition of child, maternal/family and contextual level characteristics to the empty model explained approximately 60.7% of the variability in the early childhood development in Nigeria.

## Discussion

In many parts of SSA, including Nigeria, both FI and suboptimal child development outcomes pose substantial risks. The results presented in this paper represent, to our knowledge, one of the earliest nationally representative population-level studies to use the newly designed ECDI2030 to empirically investigate the association between FI and ECD. Additionally, this study tests a multilevel model predicting ECD based on multiple factors across socio-ecological systems.

Approximately 46% of children in the sample were developmentally off track. This estimate is higher than the 37.9% prevalence of developmental delay among children in Nigeria reported in a previous study [33]. Our estimate appears to be consistent with estimates reported in previous studies. For instance, a recent multi-country study using data from the Demographic and Health Survey across 9 low and middle-income countries, which measured ECD using the original ECDI, found that the prevalence of children who were developmentally off track ranged from 7% in the Maldives to 59% in Burundi [10]. Also, a recent study showed that at least one in four children aged 3–4 years in Bangladesh were not developmentally on track [34, 35].

Notably, we found that microsystem, mesosystem and exosystem level factors play an important role in explaining the variations in ECD in children 24–59 months in Nigeria. Our results indicate one-fifth of the variation in ECD was attributable to exosystem-level factors.

This variation, although slightly lower, is fairly consistent with that reported in a previous population-based study in Nigeria showing significant clustering at the state level, with nearly one-third (29%) of the variation in ECD accounted for by differences across states [36]. Similarly, a study in Nepal showed that approximately 19% of the variation in ECD status was due to systematic differences between communities [37]. It is important to note that the variations in the prevalence estimates and the degree of geographic clustering in ECD outcome measures observed in these studies, compared with the results of our analysis, are likely attributable to differences in the analytic sample (i.e., children aged 36–59 months) and the measures of ECDI employed. Nevertheless, these findings align with Bronfenbrenner's EST, which posits that proximal contexts, within which individuals' behaviors are nested, play a crucial role in determining outcomes [20, 38]. Therefore, it is essential to critically consider the role of context in designing policies and interventions aimed at improving ECD outcomes in Nigeria.

While the effect estimates from the unadjusted model showed that children aged 24–59 months living in households experiencing moderate and severe FI were more likely to have higher odds of being developmentally off track. Controlling for all other multilevel factors, we found a null association between FI and ECD. A previous study among children <36 months in Brazil also found no association between FI and ECD [39]. Although these findings contrast with the results of other studies, several conceptual and methodological limitations in extant literature could explain the disparities in the results. These limitations include variations in the study design, composition and size of the analytic sample, methods of measuring and defining FI and ECD, data structure, analytical approaches and whether data were collected at the population or individual level. For example, a recent study in Ecuador reported strong and significantly higher associations between marginal and moderate-severe FI and global (overall) developmental delay among children age 36–59 months [9]. Several studies in Bangladesh, Ghana and Kenya employing longitudinal study designs have also reported statistically significant associations between FI and children's development [13, 40, 41]. However, the diversity of the study samples in terms of age, as well as the inclusion of varied aspects/domains of child development not included in commonly used population based surveys in these prior studies limits the extent to which their findings can be directly compared to the results of our study.

A plausible explanation for the lack of an association between FI and ECD could be attributed to variations in the measures of ECDI used in the present study. Compared to our study which used the newer ECDI2030 measure of ECD, researchers in previous studies using data from population-based surveys employed the ECDI comprising 10-items across four domains. The ECDI2030 expanded the number of items in the original ECDI, thereby capturing additional constructs across different, albeit interrelated, domains from the original ECDI. This observation has been noted in the study by Benedict and colleagues [10]. Furthermore, opinions are divided regarding the accurate definition and timing of early childhood development [42].

Another reason could be that measures of FI may not adequately capture the extent of FI or the level of hunger experienced by children, potentially leading to an overestimation of FI among this demographic and consequently, a null association with ECD and ultimately misleading inferences. Moreover, FI is measured as a household condition while hunger is an individual experience [16]. Therefore, while children experiencing hunger are likely to be food insecure, not all children living in food insecure households experience hunger [16]. Indeed, within food insecure households, parents are also likely to shield children from experiencing FI [43]. Therefore, it remains a concern whether emphasis should be on household or individual level of food deprivation. Regardless, there is need for valid and reliable instruments that better capture food deprivation during childhood, as these will further our understanding of which better predicts ECD outcomes especially in SSA.

Our results also depict a relatively homogeneous sample, with a substantial portion experiencing moderate or severe FI. This homogeneity may contribute to the lack of association observed between FI and ECD and thereby reflecting the high prevalence rates of FI across Nigeria. Furthermore, it is plausible that the impact of FI on ECD may vary depending on the child's age. For younger children, the effect of past-year FI on ECD might be evolving, whereas for older children, past-year FI may not adequately capture deviations in ECD. Hence, age-related differences in the timing and measurement of FI could influence its association with ECD outcomes.

Shifting our focus to factors associated with ECD, our results indicate that several factors across within and across multiple level of the socio-ecological model were independently associated with ECD among this population. At the microsystem level, our results showed significant positive associations of being developmentally off track with increasing child's age and those with functional difficulty. This finding is consistent with other studies conducted in different contexts. For example, a study among children in Ghana, Costa Rica and Bangladesh found that older children were less likely to achieve their developmental potential compared to their younger counterparts [44]. These findings are in contrast with the results in published studies showing children being developmentally on track with increasing age [45]. This is puzzling as evidence suggests that brain development occurs with increasing age [45]. It is important to recognize that children who report functional difficulties might also be categorized as developmentally off-track due to motor skill delays, which could affect how your results are interpreted. A recent study, however, showed children aged 4–5 years old in the UK were less likely to achieve their development potential in 2021 compared to the pre-pandemic period [46]. It is therefore likely that the disruptions associated with the COVID-19 pandemic could have had unintended consequences for ECD outcomes in Nigeria. Interestingly, our findings did not reveal sex differences in ECD and this has also been reported in a previous study [13].

At the mesosystem level our results indicate increasing levels of maternal education attainment as a significant predictor of ECD. Similar findings have been documented prior studies literature [32, 47–49]. One hypothesized mechanism explaining the relationship between maternal education and ECD, based findings from a study in Uganda, suggests that increase in the years of maternal education is likely to improve investments and engagement of mothers in stimulating activities, thereby reducing their application of harsh corporal punishment and non-home discharge [47]. Furthermore, another study which examined the differential effect of paternal versus maternal education on ECD showed that, after adjusting for microsystem and mesosytem level factors, both maternal and paternal education were positively and significantly associated with children's ECDI scores through their personal or partners efforts to support children's early learning [50]. Moreover, maternal education may exert a synergistic effect with household wealth index to improve ECD, as evidence suggests that wealthier and well educated mothers were more likely to seek early engagement of their children in early child education programs, which in turn is associated with improved ECD [51].

We observed that children in households affiliated with religions other than Christianity were less likely to be developmentally on track. Evidence comparing differences in ECD by religious affiliation is lacking. However, it has been argued that children, particularly those from Muslim-affiliated households, may experience dietary restrictions during their early years and may also be exposed to fasting in utero during periods of Holy observances. These factors, coupled with women's limited autonomy and control of household resources, may contribute to the high rate of childhood malnutrition [52, 53], consequently adversely impacting ECD. Furthermore, a study among children in India revealed that Christian children under 5 years of age were less likely to be stunted relative to their non-Christian counterparts,

an effect which was more evident among girls than for boys [54]. Further analysis would be needed to understand the underlying reasons underpinning the effect of religion and ECD.

A major contribution of this study is also the simultaneous examination of broader contextual (neighborhood) influences on ECD. We also found that neighborhood factors such as living in households in urban areas appear to be protective against developmental delays in early childhood. Similar findings which have also been echoed in prior studies documenting that residing rural residence was associated with lower ECD outcomes in Nigeria [36], Ghana [55], Vietnam [49] and China [56]. Furthermore, the reasons behind why residing in the Northern region is associated with higher odds of being developmentally off track are not fully understood, as reported also by another study in Ghana [57]. However, it is possible that subnational disparities in early childhood education and care could explain this finding. More broadly, it could also reflect regional variations in unmet need (in terms of accessibility and affordability) for programs that could enhance development in early childhood. These stated mechanisms are, however, at best, speculative and therefore warrant further investigation.

Taken together, the findings from our study underscores the complex interaction of micro-, meso- and macro-level factors which shape ECD in Nigeria and again, are consistent with the propositions of the Bronfenbrenner's Ecological Systems Theory. As studies examining development in early childhood using the SDG ECDI2030 are currently lacking globally, the results presented in this paper signal an important contribution to the existing but scant body of knowledge. By elucidating the role of several factors across multiple levels of influence on ECD, we extend current knowledge that could help address disparities in ECD outcomes for children in Nigeria.

## Strengths and limitations

The multilevel approach employed in our investigation enabled us to simultaneously examine factors across multiple levels of influence that could predict the likelihood of children being developmentally off track. Furthermore, our statistical approach allowed us to address the non-independence and inherent clustering in our data, thereby avoiding the generation of spurious estimates. By doing so, we were able to disaggregate the contextual and compositional determinants of ECD, which otherwise would not have been accounted for in more conventional single-level approaches. Additionally, our study benefited from a large sample size, providing substantial statistical power to detect group-level differences. With exposure and outcome variables ascertained using standardized and well-validated measures, our analysis offers reliable estimates that are robust and can be generalized to the population of children aged 24–59 months in Nigeria.

Several limitations threaten the validity of our results. We did not examine whether and to what extent FI was associated with the individual ECDI2030 domains. Therefore, whether the inferences drawn from our study can be extended to the association between FI and the learning, health and psychosocial well-being domains of the ECDI2030 remains open for future research. Relatedly, the ECDI2030, though measuring an array of constructs, is limited to three domains; Therefore, the findings may not be generalizable to other facets of child development not captured by this measure. Our inability to account for unmeasured variables in our analysis could have confounded our estimates of the association between FI and ECD, potentially influencing the inferences drawn from our study. Additionally, it is important to note that child anthropometric measures were expunged during the MICS6, therefore we lacked data on important variables such as stunting, wasting, and undernutrition, which could have improved the robustness of our findings. Furthermore, as our study's results are derived from population-level data, the extent to which our findings may be valid and useful in clinical

and community settings remains uncertain. Considering the cross-sectional nature of our data, we are unable to account for the effect of timing and duration of FI on ECD. Therefore, together with the exploratory approach employed to identify ecological correlates of ECD, it is important to interpret the effect estimates from the model outputs as mutually adjusted associations rather than as causal relationships to avoid the "*Table 2 fallacy*" [58].

## Implications for public health

Although the dichotomous measure of ECDI2030 enables comparability with the SDG benchmark, future research needs to explore the varied effects of FI on the multidimensional domains of ECD. These domains, as well as the overall ECDI2030 scores, are measured on a continuous scale. It may also be worthwhile to determine whether the vector means of ECD outcome domains vary based on the severity of FI, thereby providing insights into the potential impact of FI on various aspects of early childhood development.

There is an urgent need to create age-appropriate, multilevel, multicomponent interventions aimed at enhancing ECD in Nigeria. Given the evidence of geographical clustering, it is essential for the design and implementation of such interventions to consider both generalized and specific contextual effects. Further, our results highlight the need for early clinical evaluation and interventions to support the physical health of children in Nigeria who are at risk of functional limitation. This approach ensures the effectiveness of interventions targeted toward improving ECD.

Furthermore, there is a need for the evaluation of innovative approaches to screen for and address FI within clinical and community settings in children who are at risk of developmental delays. Despite the null association between FI and ECD in our study, FI is intricately related to inadequate dietary quantity and quality, which could influence child growth and development. Therefore, urgent national policy interventions, such as nutrition and early childhood development programs and social protection programs, as well as collaboration and partnerships between the private and public sectors, are needed to enable families to meet their food needs and effectively address food insecurity. Given that efforts to improve early childhood development require a multisectoral approach, there is a need for an interministerial committee to coordinate and facilitate national efforts that would employ data-driven approaches to ensure progress toward meeting the SDG target on ECD.

## Conclusion

A considerable proportion of children aged 24–59 months in Nigeria were found to be developmentally off-track. Our study, however, found no associations between ECD, as measured by the ECDI2030, and moderate or severe FI among children in this age group. Despite this, several factors across multiple levels of the socio-ecological system were identified as being associated with ECD, either as protective or risk factors, highlighting that ECD is influenced by both compositional and contextual factors. Notably, the probability of children being developmentally off-track varied across communities. Given that ECDI2030 is a relatively new measure of ECD, there is a clear need for further research to validate the results of our analysis and explore additional variables that may impact child development. To mitigate the negative consequences of suboptimal early childhood development on subsequent health, cognitive, and behavioral outcomes, efforts to address delays in ECD should focus on designing and implementing multilevel, multicomponent interventions that account for both generalized and specific contextual influences, including the broader socio-economic factors such as food insecurity.

## Supporting information

**S1 Checklist. STROBE-checklist-v4-combined-PlosMedicine.docx: STROBE checklist.**
(DOCX)

## Acknowledgments

The authors sincerely appreciate PMA for making available the data used for this study.

## Author Contributions

**Conceptualization:** Otobo I. Ujah, Omojo C. Adaji.

**Data curation:** Otobo I. Ujah.

**Formal analysis:** Otobo I. Ujah, Russell S. Kirby.

**Investigation:** Otobo I. Ujah.

**Methodology:** Otobo I. Ujah, Omojo C. Adaji, Innocent A. O. Ujah, Russell S. Kirby.

**Software:** Otobo I. Ujah.

**Supervision:** Innocent A. O. Ujah, Russell S. Kirby.

**Validation:** Innocent A. O. Ujah, Russell S. Kirby.

**Visualization:** Otobo I. Ujah.

**Writing – original draft:** Otobo I. Ujah, Omojo C. Adaji.

**Writing – review & editing:** Innocent A. O. Ujah, Russell S. Kirby.

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
