## [Decision Letter · Decision Letter 0]

25 Jun 2024

PONE-D-24-13325Food Insecurity, the Social Determinants of Health Inequities and Early Childhood Development among Children 24-59 months in Nigeria: A Multilevel AnalysisPLOS ONE

Dear Dr. Ujah,

Thank you for submitting your manuscript to PLOS ONE. After careful consideration, we feel that it has merit but does not fully meet PLOS ONE’s publication criteria as it currently stands. Therefore, we invite you to submit a revised version of the manuscript that addresses the points raised during the review process.

We look forward to receiving your revised manuscript.

Kind regards,

Mohammad Nayeem Hasan

Academic Editor

PLOS ONE

Journal Requirements: 

Reviewers' comments:

Reviewer's Responses to Questions

**Comments to the Author**

1. Is the manuscript technically sound, and do the data support the conclusions?

Reviewer #1: Yes

2. Has the statistical analysis been performed appropriately and rigorously? 

Reviewer #1: Yes

3. Have the authors made all data underlying the findings in their manuscript fully available?

Reviewer #1: Yes

4. Is the manuscript presented in an intelligible fashion and written in standard English?

Reviewer #1: Yes

5. Review Comments to the Author

Reviewer #1: Food Insecurity, the Social Determinants of Health Inequities and Early Childhood Development among Children 24-59 months in Nigeria: A Multilevel Analysis

General comment

This is a very important manuscript that addresses an important aspect of human health and development i.e. the association between food insecurity and child development at an early age. It is equally noteworthy that authors have incorporated a theoretical framework to better explain this association and determinants. Despite the fact that the manuscript is well thought-through and carried out, especially the analysis section, it still suffers from some minor errors.

Consider addressing these comments.

Keywords

Kindly consider spelling out FIES

Introduction

1. Won't it be ideal to use only HFI or only FI rather than using them interchangeably? Please check and correct if necessary.

2. Please the date for publication for reference [12] is rather 2016 and not 2010. Please update reference.

3. For this statement “The plausible mechanisms underlying the association between FI and ECD have been described by Aurino and colleagues [13].” It would have been better to rather provide a statement that describes the plausible mechanism underlining the association between FI and ECD. Kindly consider revising it.

4. The authors did not adequately address why their research centred on children in the sub-group 24 to 59 months. Either ECD includes only children within this age bracket or should provide reason(s) why this age group is important/relevant within the context of FI and ECD.

Research questions

1. Kindly correct sentence number 2.

Study design and data source

1. Kindly state that the study was cross-sectional.

Ethical considerations

1. “This study was considered not human participants research and therefore did not require institutional review board review.”

Kindly reframe to reflect that although this study is a human participant research, IRB review and approval may not be required because this present study is an analysis of secondary data. And prior ethical approval had been acquired by the primary data collectors [citation].

Figure 1

1. Please the numbers do not add up. The mistake starts from the third box. Please check and correct.

Outcome assessment

1. If the tool was developed for children aged 2 to 4 years, why is the age group in this study 24 to 59 months?

Exposure variable

1. Correct Table 1 to Table 2.

Confounding variables

1. Wouldn’t it be ideal to state that these confounding variables were based on the 4 levels of the EST framework? Or at least 3 levels since the MICs may not contain data on government policies and legislation?

This is particularly relevant as authors stated in the theoretical framework section that EST “… framework will guide our analytic strategy, the interpretation of our findings and inform how … ”

Table 2 narration

“When comparing across early child development categories, children who were not developmentally on track were more likely to be older, face functional difficulties, have mothers with less than a higher level of education, reside in households with two or more children under 5 years of age, be affiliated with a religion other than Christianity, reside in poor households, and live in rural areas and in the Northern region (p < 0.001; Table 1).”

1. With regards to the above statement, authors may have to present statistics and corresponding p-values for each of the associations stated above. E.g. “… children who were not developmentally on track were more likely to be older (52.2% versus 31.2%, p < 0.001), face functional difficulties (3.8% versus 1.6%, p < 0.001), etc

2. Add “as compared to those who are on track” to the end of the paragraph.

Table 2

1. What is the basis for providing unweighted frequencies with weighted percentages? Won’t weighted results throughout be appropriate? Even more so, the reviewer is not able to identify weighted results from unweighted results in the subsequent rows and columns.

2. When updating make sure all the numbers add up.

3. The mean age is not in the Table.

4. In most places, authors have referred to table 2 as table 1.

5. Please write out in full the classification of the wealth index.

6. Would it be better to present mean and standard deviation in this format (3.6 ± 0.25)?

Statistical analysis

1. Once a phrase has been abbreviated do not spell it out in subsequent occurrence. E.g. PVC.

Conclusion

Although the reviewer has read through the entire manuscript further comments are no longer possible due to lack of line numbers. Which makes it almost impossible to make specific references to exact phrases.

However, the manuscript has merit and provides critical information on aspects of scientific literature that have not been extensively explored, particularly in developing countries.

Kindly address these preliminary concerns.

Good luck.

6. PLOS authors have the option to publish the peer review history of their article (what does this mean?). If published, this will include your full peer review and any attached files.

Reviewer #1: **Yes: **Addae Yaw Hammond

---

## [Author Response · Author response to Decision Letter 0]

12 Aug 2024

Reviewer #1: Food Insecurity, the Social Determinants of Health Inequities and Early Childhood Development among Children 24-59 months in Nigeria: A Multilevel Analysis

General comment

This is a very important manuscript that addresses an important aspect of human health and development i.e. the association between food insecurity and child development at an early age. It is equally noteworthy that authors have incorporated a theoretical framework to better explain this association and determinants. Despite the fact that the manuscript is well thought-through and carried out, especially the analysis section, it still suffers from some minor errors.

Consider addressing these comments.

Reviewer’s comments

Keywords

Kindly consider spelling out FIES

Author’s response: Done

Reviewer’s comments

Introduction

1. Won't it be ideal to use only HFI or only FI rather than using them interchangeably? Please check and correct if necessary.

Author’s response: We have used FI throughout to maintain consistency

Reviewer’s comments

2. Please the date for publication for reference [12] is rather 2016 and not 2010. Please update reference.

Author’s response: Please note that although the paper by Lu et al. (reference 12) was published in 2016, the data cited in our introduction is from 2010. However, we have revised this sentence to better convey the central idea (Page 3, line 54).

Reviewer’s comments

3. For this statement “The plausible mechanisms underlying the association between FI and ECD have been described by Aurino and colleagues [13].” It would have been better to rather provide a statement that describes the plausible mechanism underlining the association between FI and ECD. Kindly consider revising it.

Author’s response: In the revised manuscript, we have expanded the introduction section to provide the interconnected pathways linking FI and ECD based on the points emphasized by Aurino and colleagues. (Page 3, line 59-66)

Reviewer’s comments

4. The authors did not adequately address why their research centred on children in the sub-group 24 to 59 months. Either ECD includes only children within this age bracket or should provide reason(s) why this age group is important/relevant within the context of FI and ECD.

Author’s response: We have provided an explanation for limiting our study to children 24-59 months in the revised manuscript (Page 4, line 89-92).

Reviewer’s comments

Research questions

1. Kindly correct sentence number 2.

Author’s response: Corrected (Page 4, line 81-82).

Reviewer’s comments

 Study design and data source

1. Kindly state that the study was cross-sectional (Page 5, line 140).

Author’s response: We have stated this in the revised manuscript (Page 5, line 114)

Reviewer’s comments 

Ethical considerations

1. “This study was considered not human participants research and therefore did not require institutional review board review.”

Kindly reframe to reflect that although this study is a human participant research, IRB review and approval may not be required because this present study is an analysis of secondary data. And prior ethical approval had been acquired by the primary data collectors [citation].

Author’s response: We have revised this (Page 6, line 135-138).

Reviewer’s comments 

Figure 1

1. Please the numbers do not add up. The mistake starts from the third box. Please check and correct.

Author’s response: Thank you for this observation. We have detected the error and revised the manuscript accordingly.

Reviewer’s comments 

 Outcome assessment

1. If the tool was developed for children aged 2 to 4 years, why is the age group in this study 24 to 59 months?

Author’s response: We have changed this to 24-59 months in the revised manuscript (Page 9, line 174) as contained in page 17 of THE EARLY CHILDHOOD DEVELOPMENT INDEX 2030: A NEW MEASURE OF EARLY CHILDHOOD DEVELOPMENT (https://data.unicef.org/wp-content/uploads/2023/09/ECDI2030_Technical_Manual_Sept_2023.pdf)

Reviewer’s comments 

Exposure variable: Correct Table 1 to Table 2.

Author’s response: Corrected (Page 11, line 201)

Reviewer’s comments 

Confounding variables

1. Wouldn’t it be ideal to state that these confounding variables were based on the 4 levels of the EST framework? Or at least 3 levels since the MICs may not contain data on government policies and legislation?

This is particularly relevant as authors stated in the theoretical framework section that EST “… framework will guide our analytic strategy, the interpretation of our findings and inform how … ”

Author’s response: We have revised this (Page 13, line 211-222).

Reviewer’s comments 

Table 2 narration

“When comparing across early child development categories, children who were not developmentally on track were more likely to be older, face functional difficulties, have mothers with less than a higher level of education, reside in households with two or more children under 5 years of age, be affiliated with a religion other than Christianity, reside in poor households, and live in rural areas and in the Northern region (p < 0.001; Table 1).”

With regards to the above statement, authors may have to present statistics and corresponding p-values for each of the associations stated above. E.g. “… children who were not developmentally on track were more likely to be older (52.2% versus 31.2%, p < 0.001), face functional difficulties (3.8% versus 1.6%, p < 0.001), etc

2. Add “as compared to those who are on track” to the end of the paragraph.

Author’s response: We have revised this (Page 16, line 300-309).

Reviewer’s comments 

Table 2

1. What is the basis for providing unweighted frequencies with weighted percentages? Won’t weighted results throughout be appropriate? Even more so, the reviewer is not able to identify weighted results from unweighted results in the subsequent rows and columns.

Author’s response: We avoid presenting weighted observations/frequencies to prevent any potential misrepresentation. Instead, we prefer to present the actual sample size alongside weighted percentages to maintain transparency and clarity. Moreover, the footnotes have been appropriately provided in the table describing which results are unweighted and weighted.

As an example, see the following papers: 

a. Pietrzak, R. H., Tsai, J., & Southwick, S. M. (2021). Association of symptoms of posttraumatic stress disorder with posttraumatic psychological growth among US veterans during the COVID-19 pandemic. JAMA network open, 4(4), e214972-e214972.

b. Hahn, H., Burkitt, K. H., Kauth, M. R., Shipherd, J. C., & Blosnich, J. R. (2023). Primary sources of health care among LGBTQ+ veterans: Findings from the Behavioral Risk Factor Surveillance System. Health Services Research, 58(2), 392-401.

2. When updating make sure all the numbers add up.

Author’s response: We have reviewed the numbers in the Table to ensure that all the numbers add up.

The mean age is not in the Table

Author’s response: The mean age for the sample has now been included in the revised manuscript.

4. In most places, authors have referred to table 2 as table 1.

Author’s response: Thank you for this observation. We have corrected this throughout the entire manuscript.

5. Please write out in full the classification of the wealth index.

Author’s response: Done.

6. Would it be better to present mean and standard deviation in this format (3.6 ± 0.25)?

Author’s response: We have presented this in the suggested format in the revised manuscript.

Reviewer’s comments

Statistical analysis

1. Once a phrase has been abbreviated do not spell it out in subsequent occurrence. E.g. PVC.

Author’s response: We have revised this (Page 14, line 255).

Conclusion

Although the reviewer has read through the entire manuscript further comments are no longer possible due to lack of line numbers. Which makes it almost impossible to make specific references to exact phrases. However, the manuscript has merit and provides critical information on aspects of scientific literature that have not been extensively explored, particularly in developing countries.

Author’s response: We acknowledge that errors exist across several aspects of our manuscript and have effected corrections where identified.

---

## [Decision Letter · Decision Letter 1]

17 Nov 2024

PONE-D-24-13325R1Food Insecurity and Early Childhood Development among Children 24-59 months in Nigeria: A Multilevel Mixed Effects Modelling of the Social Determinants of Health InequitiesPLOS ONE

Dear Dr. Ujah,

Thank you for submitting your manuscript to PLOS ONE. After careful consideration, we feel that it has merit but does not fully meet PLOS ONE’s publication criteria as it currently stands. Therefore, we invite you to submit a revised version of the manuscript that addresses the points raised during the review process.

We look forward to receiving your revised manuscript.

Kind regards,

Vinay Kumar, Ph.D.

Academic Editor

PLOS ONE

Journal Requirements:

**Additional Editor Comments:**

This manuscript "Food Insecurity and Early Childhood Development among Children 24-59 months in Nigeria: A Multilevel Mixed Effects Modelling of the Social Determinants of Health Inequities" needs changes as per reviewers suggestions

Reviewers' comments:

Reviewer's Responses to Questions

**Comments to the Author**

1. If the authors have adequately addressed your comments raised in a previous round of review and you feel that this manuscript is now acceptable for publication, you may indicate that here to bypass the “Comments to the Author” section, enter your conflict of interest statement in the “Confidential to Editor” section, and submit your "Accept" recommendation.

Reviewer #1: All comments have been addressed

Reviewer #2: (No Response)

2. Is the manuscript technically sound, and do the data support the conclusions?

Reviewer #1: Yes

Reviewer #2: Yes

3. Has the statistical analysis been performed appropriately and rigorously? 

Reviewer #1: Yes

Reviewer #2: Yes

4. Have the authors made all data underlying the findings in their manuscript fully available?

Reviewer #1: Yes

Reviewer #2: Yes

5. Is the manuscript presented in an intelligible fashion and written in standard English?

Reviewer #1: Yes

Reviewer #2: No

6. Review Comments to the Author

Reviewer #1: The reviewer has provided responses and where appropriate have effected changes in the manuscript to reflect reviewers comments.

Reviewer #2: Congratulations to the authors on this important contribution to the knowledge base on FI, ECD, and the applications of the socioecological model. You’ve produced a very strong analysis, thorough manuscript, and a strong addition to the literature. I particularly applaud the balance of clarity and precision in describing the models (Methods section), and the richness of the Discussion which manages to tie many broad topics together coherently. Best of luck and I look forward to reading future studies by the authors.

My specific comments are below for your consideration, some of which imply edits and others which are only to note:

Overall, this manuscript would benefit from a final review by an English-speaking editor. There are minor spelling/grammatical errors, most often in missing words and inconsistency in capitalization and spelling of words used multiple times (e.g., socioecological). But, more notably, copyediting would be useful to ensure that ideas and connecting words are clear. In a number of sentences throughout, and quite notably in the Abstract (e.g., lines 14-15, 29-31, 36), the language used is not effective in communicating the idea.

Line 21 (Abstract): Could you consider replacing the unweighted N with the weighted N, as the latter is reported in the Analytic Sample section and tables and is perhaps more useful to readers?

Throughout: Please consider adding commas in numbers >3 digits long. Tiny thing, but helps avoid errors and misinterpretations.

Lines 47-50 (Introduction): These two sentences seem not to support, but to contradict each other. Revisit to verify that “very low FI” does correspond with “lower odds” as mentioned (line 49).

Lines 81-96 (Objectives): Are these objectives listed in descending order of importance? When reading the objectives, I interpret that the main objective of the study is to answer objective 1, whereas the title, Abstract, and Introduction have been directed towards objective 2 and objective 1 has not been referenced. Consider adjusting the title to reflect the key role of the referred socioecological factors, or conversely, reordering the objectives to highlight objective 2.

Line 174 (Outcome Assessment): Kindly correct typo “24 to 59 years”.

Lines 202-203 (Exposure Assessment): Can you confirm that FIES scores were calculated only through the simple sum of affirmative responses, and that the complex assessment criteria recommended in ideal circumstances by the FAO with calculation of infit/outfit/etc. were not applied? Just confirming understanding.

Line 232 (Multilevel Model Building Strategy): Can you please define “community”, according to the MICS? Exosystem variables mentioned above include area type (urban/rural) and region, but I haven’t found a clarification of whether “communities” according to MICS correspond to one of these, or other admin boundaries, or MICS-defined clusters, or something else. Also to note for lines 423-425.

Line 271: Slightly more than one half of who?

Line 290: To clarify: severe FI only, or the exposure variable of mod-sev FI? Related in Table 2, I’m surprised to not see the aggregate numbers for mod-sev FI since this category is the key independent variable. If it’s a lot of trouble to add then no need, but at minimum a reference in the texts seems appropriate.

Line 392-394: To consider for the discussion – overlap is high in the variables being measured by “functional difficulty” and “developmentally off-track” given the components of the ECDI2030 Health section which directly enquire about motor skills. Not a problem, but something to acknowledge.

Lines 402-408: Please revise for typos which obscure the meaning of the results.

Lines 500-518: Well-said!

Lines 635-645 (Conclusion): Considering the title of the manuscript and objectives, it is surprising to not see FI mentioned at all in the conclusions. Kindly wrap up by tying all three objectives to their conclusion.

7. PLOS authors have the option to publish the peer review history of their article (what does this mean?). If published, this will include your full peer review and any attached files.

Reviewer #1: **Yes: **Addae Yaw Hammond

Reviewer #2: No

---

## [Author Response · Author response to Decision Letter 1]

6 Dec 2024

Thank you for your review and comments which have helped to improve the quality and rigor of the work undertaken to produce this manuscript. Below, we provide a point-by-point response to the observations raised. We hope your find these useful.

Overall, this manuscript would benefit from a final review by an English-speaking editor. There are minor spelling/grammatical errors, most often in missing words and inconsistency in capitalization and spelling of words used multiple times (e.g., socioecological). But, more notably, copyediting would be useful to ensure that ideas and connecting words are clear. In a number of sentences throughout, and quite notably in the Abstract (e.g., lines 14-15, 29-31, 36), the language used is not effective in communicating the idea.

Author’s response: Done

Line 21 (Abstract): Could you consider replacing the unweighted N with the weighted N, as the latter is reported in the Analytic Sample section and tables and is perhaps more useful to readers?

Author’s response: Done

Throughout: Please consider adding commas in numbers >3 digits long. Tiny thing, but helps avoid errors and misinterpretations.

Author’s response: Done

Lines 47-50 (Introduction): These two sentences seem not to support, but to contradict each other. Revisit to verify that “very low FI” does correspond with “lower odds” as mentioned (line 49).

Author’s response: Thank you for this observation. We have now corrected this in the revised manuscript.

Lines 81-96 (Objectives): Are these objectives listed in descending order of importance? When reading the objectives, I interpret that the main objective of the study is to answer objective 1, whereas the title, Abstract, and Introduction have been directed towards objective 2 and objective 1 has not been referenced. Consider adjusting the title to reflect the key role of the referred socioecological factors, or conversely, reordering the objectives to highlight objective 2.

Author’s response: We have reordered the objectives to highlight objective 2.

Line 174 (Outcome Assessment): Kindly correct typo “24 to 59 years”.

Author’s response: Done

Lines 202-203 (Exposure Assessment): Can you confirm that FIES scores were calculated only through the simple sum of affirmative responses, and that the complex assessment criteria recommended in ideal circumstances by the FAO with calculation of infit/outfit/etc. were not applied? Just confirming understanding.

Author’s response: Yes, FIES scores were calculated through the simple sum of affirmative responses.

Line 232 (Multilevel Model Building Strategy): Can you please define “community”, according to the MICS? Exosystem variables mentioned above include area type (urban/rural) and region, but I haven’t found a clarification of whether “communities” according to MICS correspond to one of these, or other admin boundaries, or MICS-defined clusters, or something else. Also to note for lines 423-425. 

Author’s response: In the context of the MICS dataset, 'community' refers to survey clusters, which represent geographically defined enumeration areas (EAs) sampled for data collection. These clusters form the primary sampling units within the hierarchical data structure. In the revised manuscript, we have changed this to “clusters”

Line 271: Slightly more than one half of who?

Author’s response: Mothers of children included in the study.

Line 290: To clarify: severe FI only, or the exposure variable of mod-sev FI? Related in Table 2, I’m surprised to not see the aggregate numbers for mod-sev FI since this category is the key independent variable. If it’s a lot of trouble to add then no need, but at minimum a reference in the texts seems appropriate. 

Author’s response: The primary independent variable in this study is food insecurity (FI) status, which is categorized into three groups: none/mild, moderate, and severe FI. We have maintained this classification throughout the text and in the Tables to distinguish between moderate and severe FI, rather than aggregating both categories into a single "food insecurity" category.

Line 392-394: To consider for the discussion – overlap is high in the variables being measured by “functional difficulty” and “developmentally off-track” given the components of the ECDI2030 Health section which directly enquire about motor skills. Not a problem, but something to acknowledge.

Author’s response: Thank you for the suggestion. We have included this in the discussion (Line 525-528)

Lines 402-408: Please revise for typos which obscure the meaning of the results.

Author’s response: Thank you. We have corrected the typos.

Lines 500-518: Well-said!

Author’s response: Thank you.

Lines 635-645 (Conclusion): Considering the title of the manuscript and objectives, it is surprising to not see FI mentioned at all in the conclusions. Kindly wrap up by tying all three objectives to their conclusion.

Author’s response: Done

---

## [Editor Report · Decision Letter 2]

11 Dec 2024

Food Insecurity and Early Childhood Development among Children 24-59 months in Nigeria: A Multilevel Mixed Effects Modelling of the Social Determinants of Health Inequities

PONE-D-24-13325R2

Dear Dr. Ujah,

We’re pleased to inform you that your manuscript has been judged scientifically suitable for publication and will be formally accepted for publication once it meets all outstanding technical requirements.

Kind regards,

Vinay Kumar, Ph.D.

Academic Editor

PLOS ONE
---

## [Editor Report · Acceptance letter]

2 Jan 2025

PONE-D-24-13325R2 

PLOS ONE

Dear Dr. Ujah, 

I'm pleased to inform you that your manuscript has been deemed suitable for publication in PLOS ONE. Congratulations! Your manuscript is now being handed over to our production team.

Kind regards, 

on behalf of

Dr. Vinay Kumar 

Academic Editor

PLOS ONE